# Flower and pod development, grain-setting characteristics and grain yield in Chinese milk vetch (*Astragalus sinicus* L.) in response to pre-anthesis foliar application of paclobutrazol

Chunfeng Zheng[1], Chunzeng Liu[1]*, Wei Ren[2], Benyin Li[1], Yuhu Lü[3], Ziliang Pan[3], Weidong Cao[4]

1 Institute of Plant Nutrition Agricultural Resources and Environmental Sciences, Henan Academy of Agricultural Sciences, Zhengzhou, China, 2 Henan Institute of Crop Molecular Breeding, Henan Academy of Agricultural Sciences, Zhengzhou, China, 3 Institute of Plant Nutrition Agricultural Resources and Environmental Sciences, Xinyang Academy of Agricultural Sciences, Xinyang, China, 4 Institute of Agriculture Resources and Regional Planning, Chinese Academy of Agricultural Sciences, Beijing, China

* zhengliu2019@126.com

**Data Availability Statement:** All relevant data are within the manuscript and its Supporting Information files.

## Abstract

The number of grains per unit land area is the most important grain yield component in Chinese milk vetch. Flower and pod survival seem to be critical determinants of grain number, which is related to the number of fertile flowers and pods during the anthesis period. Flower and pod growth are frequently considered the key determinants to establish grain number. The objective of this study was to explore the influences of paclobutrazol on flower and pod development, grain-setting characteristics and grain yield in Chinese milk vetch under different concentrations of foliar spray and try to explore the physiological regulatory mechanisms. Field experiments were carried out during the 2017–2018 and 2018–2019 growing seasons at the Dayuzhuang experimental field. The experiment involved the Chinese milk vetch cultivar "Xinzi No. 1" and six levels of foliar application of paclobutrazol, 0, 200, 300, 400, 500, and 600 mg $L^{-1}$, in treatments CK, T1, T2, T3, T4, and T5, respectively. Foliar spray was applied once, at the squaring stage. In comparison with the CK treatment, all of the paclobutrazol treatments yielded, to various degrees, increased values of the number of inflorescences per unit area, number of pods per unit area, grain-setting rate of pods, and number of grains per pod in all six inflorescence layers, with the largest increases observed in the T3 treatment. In the T3 treatment compared with the CK treatment, from the first to sixth inflorescence layers, the number of inflorescences per unit area was increased by 34.07–58.97%, the number of pods per unit area was increased by 39.69–68.35%, the grain number per pod was increased by 44.31–53.69%, and the grain-setting rate of pods was increased by 1.84–4.89%. An analysis of yield composition revealed that the paclobutrazol spray treatment had little impact on the grain weight of Chinese milk vetch. The correlations between the concentration of paclobutrazol spray and the grain yield of Chinese milk vetch reached a significant level. Grain yield was highest at the paclobutrazol concentration of 373.10 mg/L. The inflorescence contents of gibberellic acid 3 (GA₃), indole-3-acetic acid (IAA), and abscisic acid (ABA) were reduced, whereas that of cytokinin (CTK) was

**Funding:** This study was supported by China Agriculture Research System-Green Manure (CARS-22), Excellent Youth Science and Technology Fund of Henan Academy of Agricultural Sciences (2020YQ30), and National Key Research and Development Program (2018YFD0200200). The funders had no role in the study design, data collection and analysis, decision to publish, or preparation of the manuscript.

**Competing interests:** The authors have declared that no competing interests exist.

increased, by foliar application of paclobutrazol (400 mg L$^{-1}$, T3 treatment) relative to CK treatment during the stages of flowers and pods developing into grains.

## Introduction

Chinese milk vetch, which is also known as Hong Hua Cao, Qiao Yao, and Cao Zi, belongs to the family Fabaceae and genus *Astragalus* and is a traditional green manure crop in China. It improves soil physicochemical properties, increases the number and diversity of soil microorganisms, and improves soil fertility [1–5]. With the large-scale application of green manure crops in different fields under agricultural production, Chinese milk vetch grain production has become an important part of green manure production [4, 6–8]. However, basic research on Chinese milk vetch grain production remains lacking, and research on the theory and technology supporting high-yield Chinese milk vetch grain production is scarce, contributing to low and unstable grain yields [9–11]. Therefore, the development of methods for ensuring a long-term and stable supply of Chinese milk vetch seeds for production is an important topic in green manure research. Such methods can help expand the availability and use of China's green manure resources and alleviate the environmental pollution due to chemical fertilizers [1, 8, 12, 13].

Paclobutrazol is a highly efficient, slightly toxic and less residue plant growth regulator that is easily absorbed by plant roots, stems, leaves, and grains. It was first synthesized in the 1980s. It inhibits the synthesis of endogenous gibberellin and has roles in the control of height and dwarfing, cell division and elongation, and the regulation of crop growth and development [14, 15]. Many studies have investigated the effects of paclobutrazol on grain yield in other green manure crops [16–18]. Papageorgiou et al. [19] and Khan and Tewari [20] found that foliar application of paclobutrazol can inhibit plant growth and significantly increase the number of flowers, the number of pods per inflorescence, and grain yield in *Medicago sativa*. McDaniel et al. [21] showed that in *Avena sativa*, plant height was reduced and grain yield was increased by the foliar application of paclobutrazol. Marshall and Hides [22] demonstrated that foliar application of paclobutrazol can reduce canopy height, increase the number of inflorescences, the number of flowers per inflorescence, and grain weight in *Trifolium repens* L. However, to our knowledge, the effects of foliar application of paclobutrazol on the grain-setting characteristics and grain yield in Chinese milk vetch have not yet been investigated.

Therefore, the aims of the present work were to study the effects of the pre-anthesis foliar application of paclobutrazol on grain yield and the grain-setting characteristics of different inflorescence layers positions in Chinese milk vetch, as well as to determine whether the responses are related to the hormone contents in inflorescences during the periods of flower and pod develop into grains.

## Materials and methods

### Plant materials and experimental design

This study was conducted in the field in Dayuzhuang (32˚16'N, 114˚11'E), Lanqing Township, Zhengyang County, Henan Province, China, from 2017 to 2019. The test field soil type was lime concretion black soil, and the texture was clay. The organic matter content of the 0–20 cm soil layer was 17.2 g kg$^{-1}$, the total nitrogen was 0.9 g kg$^{-1}$, the alkali-hydrolysis nitrogen was 102.98 mg kg$^{-1}$, the available phosphorus was 28.7 mg kg$^{-1}$, and the available potassium was 125.42 mg kg$^{-1}$. Before sowing, all experimental plots were fertilized with 187.5 kg ha$^{-1}$ of

compound fertilizer (N: $P_2O_5$: $K_2O$ = 24: 11: 10) as a basal application. Xinzi No. 1 was chosen as the experimental material. Seeds were sown on 15 September in both the 2017–2018 and 2018–2019 growing seasons. The plant density was 75 plants $m^{-2}$ with an equal row spacing of 25 cm. For sowing, the seeds were mixed well with fine sand and then sown. Field management generally followed the local standard production practices for high-yielding Chinese milk vetch. Foliar spray containing water alone (CK) or water with paclobutrazol at 200 mg $L^{-1}$ (T1), 300 mg $L^{-1}$ (T2), 400 mg $L^{-1}$ (T3), 500 mg $L^{-1}$ (T4), or 600 mg $L^{-1}$ (T5) was sprayed once per treatment at the squaring stage (prior to anthesis) at a volume of 750 kg $hm^{-2}$ each. Paclobutrazol was a 15% wettable powder which was commercial reagent and provided by Anyang Quanfeng Biotechnology Co., LTD. The protocol resulted in a layer of spray deposited on the leaf surfaces, without dripping. The experimental design was a completely randomized block design, with each treatment performed in triplicate. Each microcrop was an experimental unit, and each plot area was 20 $m^2$.

## Sampling and measurements

Based on the growth characteristics of Xinzi No. 1, each plant was visually divided into six inflorescence layers from bottom to top (Fig 1). The sixth inflorescence layer comprised the 6th inflorescence and all inflorescences above it (Fig 1). Three inflorescence layer positions were then defined: the apical position, comprising the fifth and sixth layers; the central position, comprising the third and fourth layers; and the basal position, comprising the first and second layers (Fig 1). The timing of critical plant growth and developmental stage proposed by Zhong (1978) was followed; squaring stage (50% plants appear flower buds), initial anthesis (20% plants flowering, *i.e.*, inflorescence appearance time), pod bearing stage (50% plants appear pods), maturity stage (80~85% pods turn yellow or yellowish brown). The samples collected were observed under a SZN-6 (45x) stereomicroscope to record (1) the flower and pods development process on the main stem; (2) the flower and pod morphological characteristics at the different developmental stages; (3) the flower and pod number; and (4) the differentiated grain number.

In early May of the following year, during the maturity stage, 1 $m^2$ of the field was randomly selected which was away from the edge of the plot. Each inflorescence layer on the main stem and lateral branches was separated and the inflorescences numbers, pods numbers, average grain numbers per pod and grain-setting percentage of pods per unit area were determined. The numbers of effective plants, primary branches, pods (both fertile pods and sterile pods), and grain numbers and grain weight were measured. The harvest from 2 $m^2$ away from the edge of the plot was used to calculate the grain yield. From these measurements, the average number of grains per pod and the grain-setting rates of the pods in the various inflorescence layers per unit area were calculated as follows:

The average number of grains per pod in an inflorescence layer = the number of grains in the inflorescence layer/the number of pods in the inflorescence layer.

The grain-setting rate of pods in an inflorescence layer (%) = the number of pods with seeds in the inflorescence layer/number of pods in the inflorescence layer × 100%.

In each treatment (CK, T1, T2, T3, T4, and T5), a 1 g sample of inflorescence from the central inflorescence layers was collected at the squaring stage, at initial anthesis and at 7, 14, 21, 28, 35, and 42 d after initial anthesis. Each sample was ground with liquid nitrogen and mixed three times with an extracting solution containing 80% methanol. The inflorescence homogenates were stored overnight at –20˚C and then centrifuged (8000 rpm, 4˚C, 1 h). The supernatants were extracted through a C-18 solid-phase extraction column (Waters Oasis MCX 6cc (150mg), Guofan Industry Co., LTD, Shanghai, China). Then, the precipitates were dried at

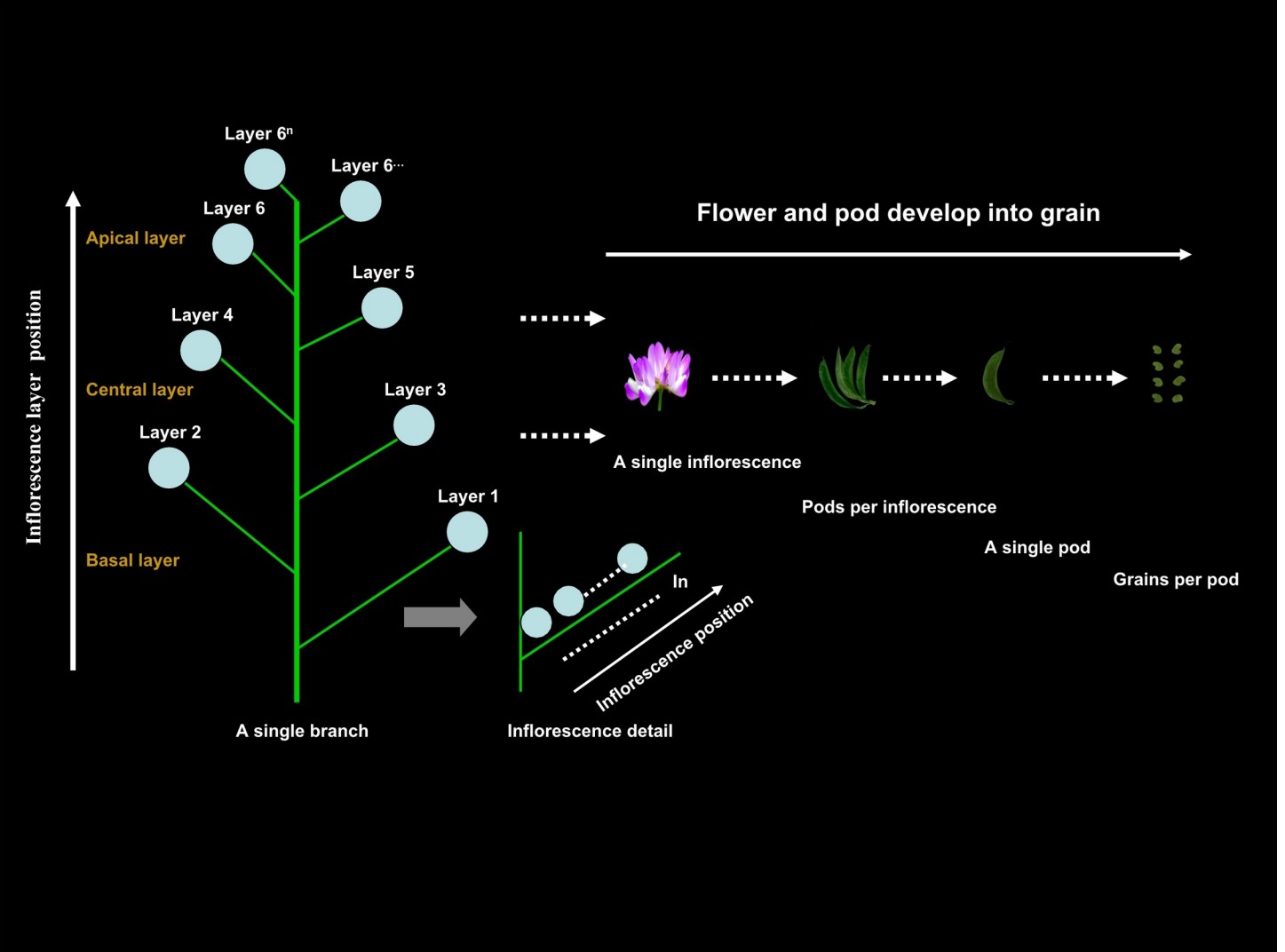

**Fig 1. Morphological structure of the Chinese milk vetch plant.** The photographs were captured using a stereomicroscope (SZN-6 (45x), OPTIKA S.R.L., ITALY). The left panel illustrates the inflorescence layers (from bottom to the top). The left bottom panel illustrates the inflorescence position of the inflorescence layers, and the inflorescences are numbered according to their positions relative to the rachis from most proximal to most distal. The right panel illustrates the processes of flower and pods develop into grain. The illustrations and photographs are not to scale; for reference, pod width and pod length are approximately 3 mm and 23 mm, respectively.

40˚C in a vacuum, homogenized with 1 ml phosphate buffer solution (PBS) (pH 7.4) buffer solution, and left to rest at room temperature for 30 min. After centrifugation (8000 rpm, 4˚C, 15 min), the supernatants were extracted and stored at -20˚C until analysis. The contents of endogenous hormones in the aliquots of the supernatants were measured by the UPLC-MS/MS method following Kang et al. [23].

## Climate variations during growth periods

The area has a warm temperate semi-humid continental monsoon climate, the climatic conditions were approximately similar between the 2017–2018 and 2018–2019 growing periods (Fig 2). In two growing seasons, a serious drought appeared in this region during the early stage of Chinese milk vetch growth. From December to the end of January, rainfall was merely 16.7 mm in 2017–2018, and rainfall was merely 17.5 mm in 2018–2019. During the early spring,

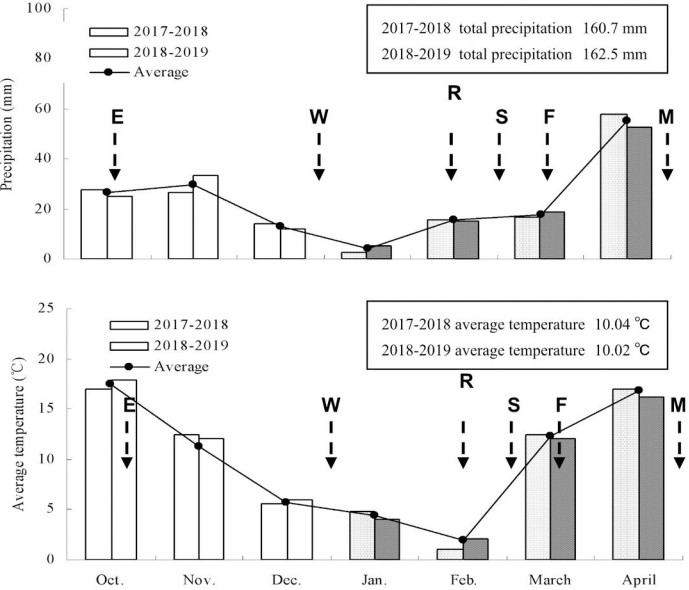

**Fig 2. Accumulated precipitation (bars), the average temperatures (bars), and the average data (points and line) over the 2017–2018 and 2018–2019 Chinese milk vetch growing periods at the experimental field (Zhengyang, China).** The arrows denote the dates: seeding emergence stage (E), wintering stage (W), returning green stage (R), squaring stage (S), anthesis stage (F), and maturity stage (M) for Chinese milk vetch.

the temperature was low, the average temperature in February was 1.03˚C and 2.02˚C in 2017–2018 and 2018–2019, respectively. During the whole growth seasons, the total accumulated temperature was 1957˚C and 1930˚C in 2017–2018 and 2018–2019, respectively, the monthly average temperature was 10.04˚C and 10.02˚C in 2017–2018 and 2018–2019, respectively, the annual total precipitation was 160.7 mm and 162.5 mm in 2017–2018 and 2018–2019, respectively (Meteorological data of 2017–2019, provided by Zhengyang Meteorological Bureau).

## Statistical analysis

All data were assessed by analysis of variance (ANOVA), performed using PASW (version 18.0), to examine differences among the spraying treatments. Least significant differences (LSDs) were calculated at a probability level of P = 0.05. Preliminary analysis (*i.e.*, Table 1) demonstrated that the results were similar between the two growing seasons; thus, the averages over the two seasons are presented in the figures. Microsoft Excel® 2003 and PASW Statistics 18.0 software were used for data analysis and post-processing.

## Results

### Effects of foliar application of paclobutrazol on yield and yield components

As expected, the foliar application of paclobutrazol resulted in increased grain yields at maturity during both the 2017–2018 and 2018–2019 growing periods, with the largest increase observed in the T3 treatment (Table 1). The grain yield per hectare and its components (pod numbers, grains per pod, and grain weight) have no significance between the 2017–2018 and 2018–2019 growing periods (Table 1). Compared with the plants treated with the water control, the grain yields of the T3-treated plants increased by 16.70% from 2017–2018 and by 18.25% from 2018–2019. The pod number and grains per pod were significantly increased in

**Table 1. Grain yield and main yield components of Chinese milk vetch in the six spraying treatments.**

| Growing period | Treatment | Pod number (×10⁴·hm⁻²) | Grains per pod | 1000-grain weight (g) | Grain yield (Kg·hm⁻²) |
|---|---|---|---|---|---|
| 2017–2018 | CK | 3540.34±30.11c | 3.42±0.07c | 2.98±0.05a | 478.50±17.02c |
| | T1 | 3558.50±25.41b | 3.90±0.10bc | 3.04±0.07a | 503.00±19.21b |
| | T2 | 3570.95±44.15ab | 4.27±0.11b | 3.06±0.10a | 516.00±15.23ab |
| | T3 | 3590.47±40.26a | 5.09±0.09a | 3.26±0.11a | 558.40±15.34a |
| | T4 | 3581.20±2 6.45ab | 4.67±0.07ab | 3.17±0.06a | 534.51±16.01ab |
| | T5 | 3573.50±33.12ab | 4.48±0.10b | 3.21±0.07a | 520.12±19.42ab |
| 2018–2019 | CK | 3542.11±45.61c | 3.49±0.12c | 3.04±0.14a | 481.60±20.02c |
| | T1 | 3559.50±37.48b | 4.01±0.08bc | 3.05±0.06a | 514.10±16.45b |
| | T2 | 3577.95±56.14ab | 4.38±0.12b | 3.17±0.05a | 527.10±17.17ab |
| | T3 | 3599.47±46.31a | 5.24±0.14a | 3.37±0.14a | 569.51±13.42a |
| | T4 | 3584.20±66.01ab | 4.78±0.06ab | 3.28±0.07a | 545.60±19.64ab |
| | T5 | 3580.50±27.15ab | 4.59±0.15b | 3.32±0.13a | 531.20±17.78ab |
| | MS (T) | 103422.12* | 154** | 196.291NS | 22541.241*** |
| | MS (Y) | 13332.4NS | 27.46NS | 84.16NS | 3345.148NS |
| | MS (T×Y) | 10046.74NS | 1.17NS | 4.499NS | 1002.473NS |

*Note*: The values are means ± standard error (± SE), n = 3. Different lowercase letters within a column indicate significant differences between treatments at p < 0.05. The mean square (MS) for the effects of treatments (T), years (Y),and their interaction (T×Y) are also shown. *, **, ***, and NS stand for the levels of significance of the MS values (0.05, 0.01, 0.001, and non-significant, respectively).

response to the T3 treatment compared with the water treatment during both growing periods; however, no differences in thousand-grain weight were noted (Table 1). The pod number per hectare and grains per pod increased by 50.13 ×10⁴ plants and 1.67 grains, respectively, in response to the T3 treatment in 2017–2018, and these values increased by 57.36 ×10⁴ plants and 1.75 grains, respectively, in 2018–2019.

Taken together, most of the large effects of foliar application of paclobutrazol on yield were due to the effects on the grain numbers (the number of pods and grains per pod). Then, to ascertain the origin of the yield responses to treatments, the focus turned to analyzing the effects of treatments on the determinants of grain numbers (*i.e.*, number of inflorescences, pods, grains per pod and grain-setting rate of pods).

## Effect of foliar application of paclobutrazol on the determinants of grain number

**Number of inflorescences in different inflorescence layers.** From the first to sixth inflorescence layers, the number of inflorescences per unit area was significantly higher for the plants that received paclobutrazol spray treatment than for the control plants, with the largest difference observed between the T3 treatment and the control (Fig 3). We found that from the first to sixth inflorescence layer, the number of inflorescences per unit area was increased in the paclobutrazol treatments compared with the control by 18.02–39.53%, 7.44–34.07%, 20.05–42.74%, 14.83–43.92%, 17.95–57.45%, and 23.08–58.97%.

**Number of pods in different inflorescence layers.** The number of pods per unit area in the various inflorescence layers was higher in all of the paclobutrazol treatments than in the control (Fig 4). Treatments T3, T4, and T5 showed the greatest differences from the control, which decreased in the order T3>T4>T5. The T1 and T2 treatments did not significantly differ from the control (Fig 4). From the first to sixth inflorescence layer, the number of pods per unit area was increased after spraying paclobutrazol at various concentrations compared with

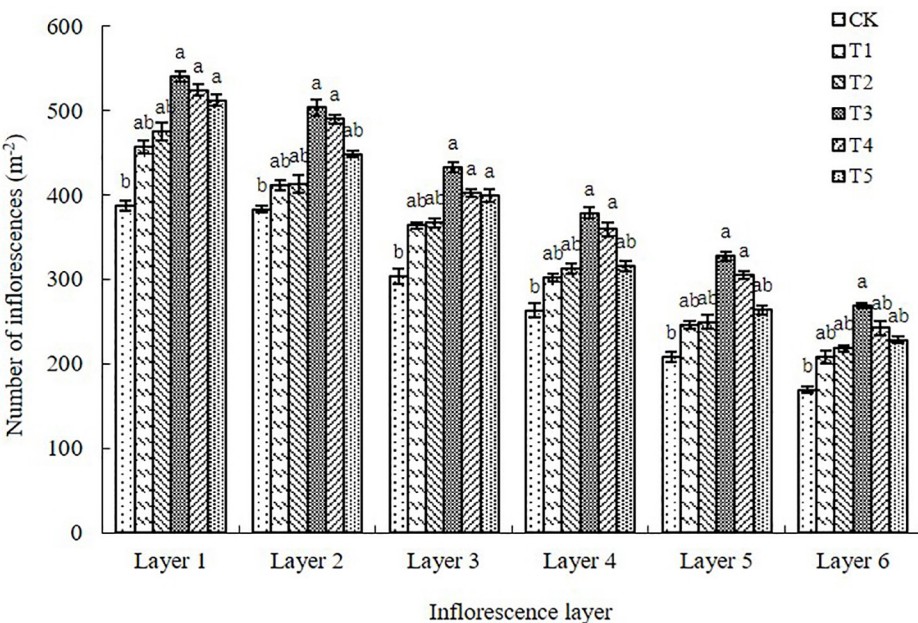

**Fig 3. Number of inflorescences per unit area for each inflorescence layer on the main stem and lateral branches of the Chinese milk vetch cultivar "Xinzi No. 1" under six foliar spray treatments (CK: Water; T1: 200 mg L$^{-1}$ paclobutrazol; T2: 300 mg L$^{-1}$ paclobutrazol; T3: 400 mg L$^{-1}$ paclobutrazol; T4: 500 mg L$^{-1}$ paclobutrazol; T5: 600 mg L$^{-1}$ paclobutrazol).** The data are presented as treatment mean±standard error (±SE), n = 3. Different lowercase letters indicate significant differences (p<0.05).

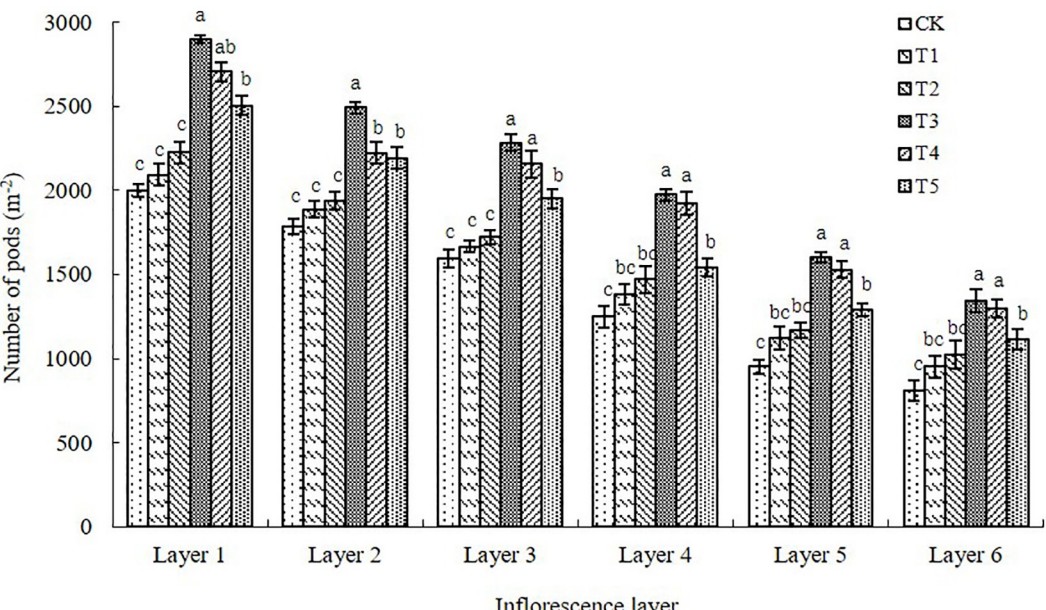

**Fig 4. Number of pods per unit area for each inflorescence layer on the main stem and lateral branches of the Chinese milk vetch cultivar "Xinzi No. 1" following six foliar spray treatments (CK: Water; T1: 200 mg L$^{-1}$ paclobutrazol; T2: 300 mg L$^{-1}$ paclobutrazol; T3: 400 mg L$^{-1}$ paclobutrazol; T4: 500 mg L$^{-1}$ paclobutrazol; T5: 600 mg L$^{-1}$ paclobutrazol).** The data are presented as treatment mean±standard error (±SE), n = 3. Different lowercase letters indicate significant differences (p<0.05).

the number in the control by 4.75–45.19%, 5.72–39.69%, 4.53–43.21%, 10.70–57.88%, 18.17–68.35%, and 17.59–65.67%.

**Number of grains per pod in different inflorescence layers.** As shown in Fig 5, the number of grains per pod in each plant inflorescence layer was higher under the paclobutrazol treatments than under the control, with the largest increase observed in the T3 treatment. No significant increase was observed in the T1 treatment relative to the control (Fig 5). From the first to sixth inflorescence layer, the number of grains per pod was increased in the paclobutrazol treatments compared with the control by 10.28–46.27%, 19.94–48.55%, 19.59–48.53%, 12.28–44.31%, 18.97–53.69%, and 18.12–51.46%.

**Grain-setting rate of pods in different inflorescence layers.** The grain-setting rates in the inflorescence layers were significantly higher after the T3, T4, and T5 paclobutrazol spray treatments than after control (Fig 6). The largest difference was between the T3 and the control, whereas the differences between T1 or T2 and the control were not significant (Fig 6). From the first to sixth inflorescence layer, the grain-setting rate of pods was greater for plants treated with paclobutrazol than for plants receiving control by 0.02–2.51%, 0.24–2.24%, 0.34–2.31%, 0.24–1.84%, 1.31–4.08%, and 1.01–4.89%.

## Relationship between paclobutrazol concentration and grain yield

Quadratic curves were found to fit the relationships between paclobutrazol concentration and grain yield in Chinese milk vetch (Fig 7). The grain yield first increased and then decreased with increasing paclobutrazol concentration, and the coefficients of the regression equation was 0.883, the correlations reached a significant level (Fig 7). According to the regression equations, the concentration of 373.10 mg $L^{-1}$ was the predicted optimal concentration, with grain yield predicted to be highest at this concentration.

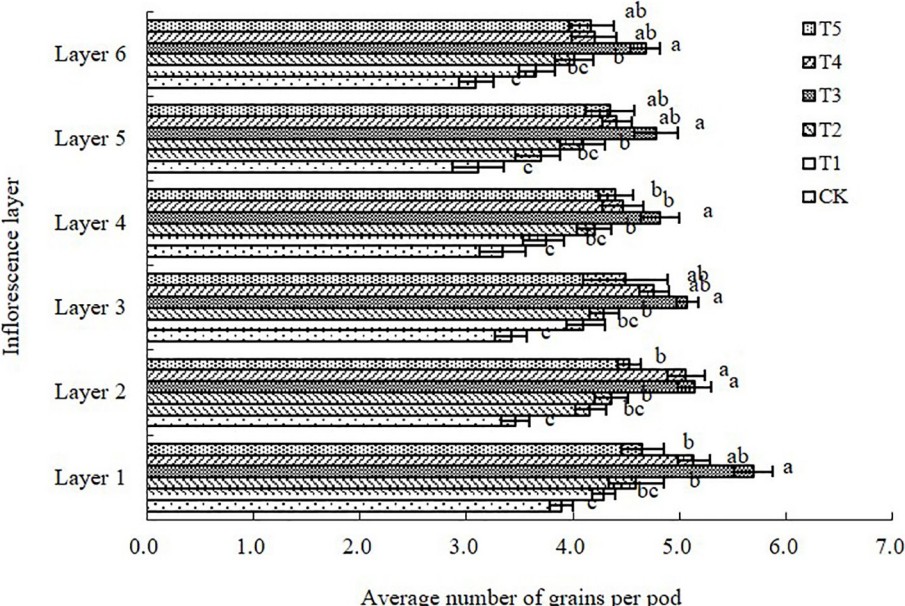

**Fig 5. Average number of grains per pod for each inflorescence layer on the main stem and lateral branches of the Chinese milk vetch cultivar "Xinzi No. 1" after six foliar spray treatments (CK: Water; T1: 200 mg $L^{-1}$ paclobutrazol; T2: 300 mg $L^{-1}$ paclobutrazol; T3: 400 mg $L^{-1}$ paclobutrazol; T4: 500 mg $L^{-1}$ paclobutrazol; T5: 600 mg $L^{-1}$ paclobutrazol).** The data are presented as treatment mean±standard error (±SE), n = 3. Different lowercase letters indicate significant differences (p<0.05).

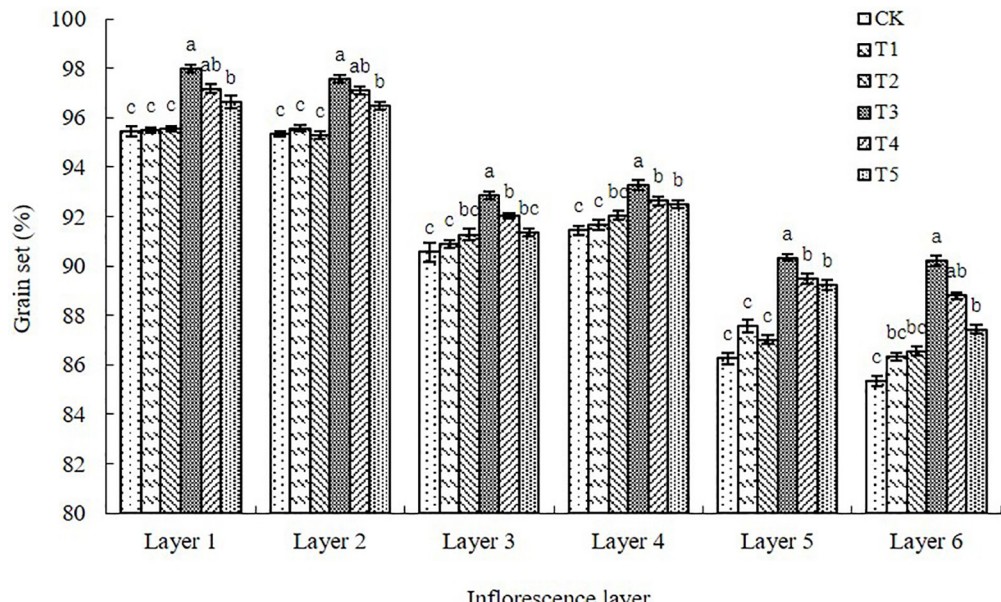

**Fig 6. Grain-setting percentage of pods for each inflorescence layer on the main stem and lateral branches of the Chinese milk vetch cultivar "Xinzi No. 1" under six foliar spray treatments (CK: Water; T1: 200 mg L$^{-1}$ paclobutrazol; T2: 300 mg L$^{-1}$ paclobutrazol; T3: 400 mg L$^{-1}$ paclobutrazol; T4: 500 mg L$^{-1}$ paclobutrazol; T5: 600 mg L$^{-1}$ paclobutrazol).** The data are presented as treatment mean±standard error (±SE), n = 3. Different lowercase letters indicate significant differences (p<0.05).

## Effects of paclobutrazol on inflorescence contents of endogenous hormones

Over the course of flower and pod development, GA$_3$ content in the inflorescences in the control and paclobutrazol-treated plants first decreased, then increased, then decreased, and then increased, with the overall trend exhibiting a "W" shape (Fig 8A). Overall, inflorescence GA$_3$ content was lower in the plants sprayed with paclobutrazol than in the control plants (Fig 8A). The GA$_3$ content in the plants sprayed with paclobutrazol was significantly lower than that in the control plants from 7 to 28 d after initial anthesis

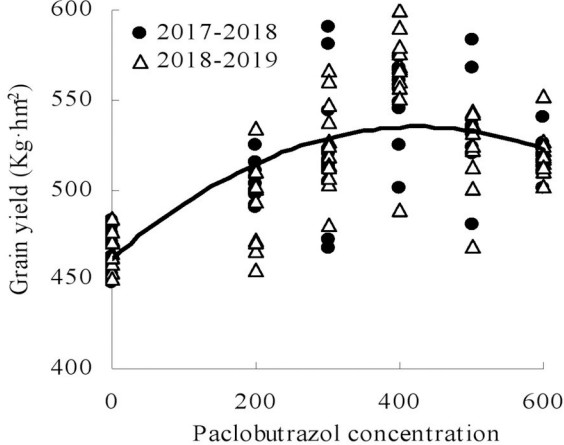

**Fig 7. Relationship between paclobutrazol concentrations and grain yield of Chinese milk vetch ($Y = -0.0005X^2 + 0.3731X+459.97$, $R^2 = 0.883$, $P<0.001$, $n = 72$).** Six foliar spray paclobutrazol concentrations during the growing periods at the bottom: 0 mg L$^{-1}$, 200 mg L$^{-1}$, 300 mg L$^{-1}$, 400 mg L$^{-1}$, 500 mg L$^{-1}$, 600 mg L$^{-1}$.

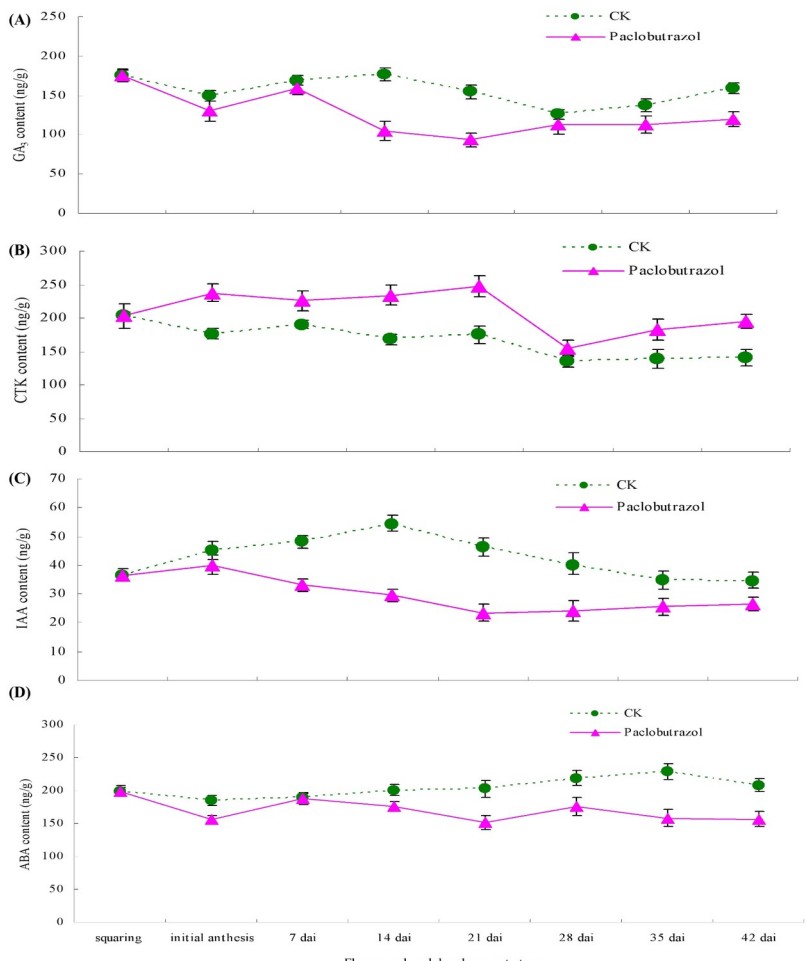

**Fig 8. The inflorescence contents of GA$_3$, CTK, IAA and ABA under two foliar spray treatments (water, paclobutrazol 400 mg L$^{-1}$) during flower development and pod development in Chinese milk vetch.** The inflorescence samples were collected at the following stages (bottom of the figure): squaring stage; initial anthesis stage; and 7, 14, 21, 28, 35, and 42 d after initial anthesis.

(Fig 8A). The GA$_3$ content in control plants was low at the initial anthesis stage and at 28 d after anthesis and peaked at 14 d after initial anthesis. The GA$_3$ content in plants with paclobutrazol treatment exhibited a low level at 21 d after initial anthesis and peaked at 7 d after initial anthesis (Fig 8A).

Inflorescence cytokinin (CTK) content was significantly higher in the plants receiving paclobutrazol than in the control plants during the development stages of flower and pods, except 28 days after initial anthesis; the content first increased, then decreased and finally increased again (Fig 8B). The inflorescence CTK content in plants receiving paclobutrazol peaked at 21 d after initial anthesis and reached its lowest level at 28 d after initial anthesis. However, the CTK content in the control plants declined slightly after the squaring stage, reaching its lowest level at 28 d after initial anthesis (Fig 8B).

Indole-3-acetic acid (IAA) content was significantly lower in the plants receiving paclobutrazol than in the controls from initial anthesis to 42 d after initial anthesis (Fig 8C). In the control plants, the inflorescence IAA content first increased and then decreased. It peaked at 14 d after initial anthesis and then rapidly decreased. In the plants receiving paclobutrazol, the

inflorescence IAA content rapidly decreased after initial anthesis, reaching its lowest level at 21 d after initial anthesis, and then began to increase slowly (Fig 8C).

Abscisic acid (ABA) content was significantly lower in the plants receiving paclobutrazol than in the control plants from 14 to 42 d after initial anthesis (Fig 8D). In the control plants, inflorescence ABA content first decreased, then increased, then decreased. It reached its lowest level at anthesis and peaked at 35 d after initial anthesis. In the plants receiving paclobutrazol, ABA content first decreased, then increased, decreased again, increased again and finally decreased, with the overall trend exhibiting a "fluctuated" shape, reaching its lowest levels at anthesis and 21 d after initial anthesis (Fig 8D).

## Discussion

### Developmental patterns of flower and pods in Chinese milk vetch

In Chinese milk vetch, inflorescences flower in sequence from bottom to top [3, 24], and pod setting occurs in this same order (Fig 1). The flowers and pods of the Chinese milk vetch are shed in this order as well. Within an inflorescence, blooming occurs from the outside to the inside, and pod setting and shedding occur in this order as well [24, 25]. More flowers are found on the central and basal inflorescences of Chinese milk vetch stems and branches than at the top. The basal inflorescence has the most flowers (Fig 1). The flower- and pod-shedding rates are determined by the flower's position on the stems and branches. The shedding rates of the flowers and pods are generally lower on the basal inflorescence than on the apical inflorescence [24]. In this study, we found that the numbers of inflorescences and pods per unit area of Chinese milk vetch gradually decreased as the inflorescence layer increased (Figs 2 and 3), which is consistent with the results of a previous study [24].

### Grain-setting characteristics of Chinese milk vetch

The anthesis period of Chinese milk vetch is long; accordingly, the flower-and pod-shedding duration is long, potentially lasting for approximately 38–43 d. Most fall off between the full-bloom stage and the end of the anthesis stage, and the flower abscission rate is nearly 85%, whereas the pod abscission rate is nearly 75% [25, 26]. In exploring flower and pod shedding by Chinese milk vetch, Zhong [24, 26] showed that Chinese milk vetch cultivation aimed at promoting grain yield should not merely emphasize plant growth, because such growth results in long stems and leaves. Increasing plant height does not increase grain yield and may reduce the pod-setting rate of the inflorescences. To promote high grain yield, the focus should be on improving the number of flowers and pods per inflorescence and the grain-setting rate [26].

In the present study, we showed that foliar application of paclobutrazol at the critical development period of flowers and pods in Chinese milk vetch increased the numbers of inflorescences and pods, the grain-setting rate of pods and the number of grains per pod in all six inflorescence layers. In particular, paclobutrazol significantly promoted grain setting in the weak inflorescence layers at the F5 and F6 positions, which can promote the formation of pods with more grains (Figs 3, 4, 5 and 6). The responses to foliar spraying were almost exclusively due to increased proportions of effective flowers and pods that successfully set grains.

### Possible factors influencing grain setting

Hormones are important factors affecting plant growth and development and are important substances in plant growth regulation. Papageorgiou et al. [19] proposed that paclobutrazol regulates rice growth and development by regulating the balance and interactions between various endogenous hormones. Pharis and King [27] found that paclobutrazol's regulatory effect

on the growth and development of *Brassica napus* was related to the contents of IAA, GA, and ethylene. Amir et al. [28] reported that the development of plant reproductive organs is related to the ratios of endogenous hormones in plants. Oliveira et al. [29] suggested that measures can be adopted during the critical development periods of flowers and pods to increase the grain yield and the grain-setting characteristics of Chinese milk vetch.

In our current study, paclobutrazol treatment was found to significantly promote grain setting in different inflorescence layers, and it especially promoted grain setting in the weak inflorescence layers in the apical position. Flower development and pod development are the result of a regulated balance among meristem size and coordination and organ initiation, and floral meristem size is regulated by cytokinins, gibberellins, ABA, and auxins, which play major roles in organ initiation and organogenesis [15, 30]. In the present study, we found that the inflorescence contents of $GA_3$, IAA, and ABA were reduced while that of CTK was increased by foliar application of paclobutrazol during the stage of flowers and pods developing into grains (Fig 8). Thus, we conclude that during the critical development period of flowers and pods, foliar application of paclobutrazol may alter the levels of endogenous hormones in inflorescences, resulting in increased flower development and grain setting and improved final grain yield. The physiological mechanisms by which paclobutrazol application regulates grain formation requires further study.

## Conclusions

This study investigated the effects of paclobutrazol treatment on grain-setting characteristics and grain yield in Chinese milk vetch cultivar "Xinzi No. 1". We demonstrated that at the critical development period of flowers and pods in Chinese milk vetch (at the squaring stage, prior to anthesis), the numbers of inflorescences and pods per unit area, the grain-setting rate of pods, and the number of grains per pod in different inflorescence layers can be significantly increased by foliar application of paclobutrazol, thereby improving final grain yield. Grain yield is expected to be highest at a paclobutrazol concentration of 373.10 mg $L^{-1}$. In addition, the foliar application of paclobutrazol improved the inflorescence content of CTK and reduced the inflorescence contents of $GA_3$, IAA, and ABA during the stage of flowers and pods developing into grains. This study showed that paclobutrazol application has the potential to increase the grain yield and grain-setting characteristics of Chinese milk vetch. Furthermore, this study confirms that flower/pod survival is a major determinant of grain number in Chinese milk vetch and that the grain-setting process seems to be mediated by the balance of hormones in inflorescences.

## Supporting information

**S1 Dataset. Relevant data underlying the findings described in this manuscript.**
(XLS)

**S2 Dataset. Relevant data underlying the findings described in this manuscript.**
(XLS)

**S3 Dataset. Relevant data underlying the findings described in this manuscript.**
(XLS)

## Acknowledgments

We thank the two anonymous reviewers for their helpful comments and supplementary proposal. We thank the native English speakers (AJE) who provided assistance with language correction.

## Author Contributions

**Data curation:** Chunfeng Zheng, Wei Ren, Benyin Li, Yuhu Lü, Ziliang Pan.

**Formal analysis:** Weidong Cao.

**Investigation:** Chunfeng Zheng.

**Methodology:** Chunfeng Zheng.

**Supervision:** Chunzeng Liu.

**Writing – original draft:** Chunfeng Zheng.

**Writing – review & editing:** Chunfeng Zheng.

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
