## [Decision Letter · Decision Letter 0]

21 Oct 2020

PONE-D-20-26606

Flower and pod development, grain-setting characteristics and grain yield in Chinese milk vetch (Astragalus sinicus L.) in response to pre-anthesis foliar application of paclobutrazol

PLOS ONE

Dear Dr. Liu,

Thank you for submitting your manuscript to PLOS ONE. After careful consideration, we feel that it has merit but does not fully meet PLOS ONE’s publication criteria as it currently stands. Therefore, we invite you to submit a revised version of the manuscript that addresses the points raised during the review process.

We look forward to receiving your revised manuscript.

Kind regards,

Vijay Gahlaut, Ph.D.

Academic Editor

PLOS ONE

Journal Requirements:

Reviewers' comments:

Reviewer's Responses to Questions

**Comments to the Author**

1. Is the manuscript technically sound, and do the data support the conclusions?

Reviewer #1: Yes

Reviewer #2: Yes

2. Has the statistical analysis been performed appropriately and rigorously? 

Reviewer #1: Yes

Reviewer #2: Yes

3. Have the authors made all data underlying the findings in their manuscript fully available?

Reviewer #1: Yes

Reviewer #2: Yes

4. Is the manuscript presented in an intelligible fashion and written in standard English?

Reviewer #1: Yes

Reviewer #2: Yes

5. Review Comments to the Author

Reviewer #1: Please check the reviewer's attachment.

Technical comments:

1. Methods of application, selection of statistical design and growth parameters studies are appropriate.

2. Mention the purity of Paclobutrazol, company name and whether it is commercial or plant cell culture tested reagent? Also mention which surfactant at what concentration was used for foliar application?

3. Since the plant is mainly used as green manure, study or data related to residual analysis of Paclobutrazol should be included. There are reports of negative effects of Paclobutrazol on soil microflora and next crop to be cultivated. (https://www.sciencedirect.com/science/article/pii/S0929139302001105?via%3Dihub;
https://link.springer.com/article/10.1007/s11356-019-05947-9)

General comments:

1. Please mention one format of unit i.e. /L or l-1 for all the unit types.

2. Sub-heading 3.3 should be corrected as other sub-headings.

Reviewer #2: The manuscript showing basic research. But representation of data and conduction of experiment is technically good. Statistical analysis is also performed appropriately. Language of manuscript is also clear and correct.

I have one query regarding to use of single cultivar with different levels of Paclobutrazol. How we can recommend the promising treatment on the basis of single cultivar?

6. PLOS authors have the option to publish the peer review history of their article (what does this mean?). If published, this will include your full peer review and any attached files.

Reviewer #1: No

Reviewer #2: No

---

## [Author Response · Author response to Decision Letter 0]

15 Dec 2020

Dear Editors and Reviewers: 

On behalf of my co-authors, we thank you very much for giving us an opportunity to revise our manuscript, we appreciate you and reviewers very much for your positive and constructive comments and suggestions on our manuscript entitled “Flower and pod development, grain-setting characteristics and grain yield in Chinese milk vetch (Astragalus sinicus L.) in response to pre-anthesis foliar application of paclobutrazol” (PONE-D-20-26606). We have studied reviewer’s comments carefully and have made revision which marked in red in the paper. We have tried our best to revise our manuscript according to the comments. We hope that the revision is acceptable, and we look forward to hearing from you soon.

Correspondence and phone calls about this paper should be directed to Chunzeng Liu at the following address, phone and e-mail:

Address: Institute of Plant Nutrition Agricultural Resources and Environmental Sciences, Henan Academy of Agricultural Sciences, #116 Huayuan Road, Zhengzhou, Henan 450002, China

Tel.: 86-371-65738534

Fax: 86-371-65738534

E-mail: zhengliu2019@126.com

Thanks very much again for your attention to our paper．

With best wishes,

Yours sincerely, 

Chunzeng Liu 

For your guidance, the responses to your comments are appended below:

Comment 1: Mention the purity of paclobutrazol, company name and whether it is commercial or plant cell culture tested reagent? Also mention which surfactant at what concentration was used for foliar application?

Response: We thank for your careful check and have accepted this suggestion. In the revised manuscript, the purity of paclobutrazol, company name and it is commercial has been described and added in detail in L111–113 page 6. The recommended concentration dosage of paclobutrazol was used for foliar spraying on crops was 50~1000 mg• kg-1, and on fruit trees was 1000~1500 mg• kg-1. 

Comment 2: Since the plant is mainly used as green manure, study or data related to residual analysis of Paclobutrazol should be included. There are reports of negative effects of paclobutrazol on soil microflora and next crop to be cultivated.

Response: We thank for your careful check. The residue of paclobutrazol with different dosage applied to different crops and soil is different. Our research group used gas chromatography-nitrogen and phosphorus detection method to detect the paclobutrazol residue in soil, plant, pod and seed of Chinese milk vetch. It was discovered that at harvest, there was no residue in soil, plant, pod and seed of Chinese milk vetch after using applying recommended dosage（200~600 mg•L-1）of 15% paclobutrazol WP once at squaring stage (prior to anthesis). Because Chinese milk vetch is one green manure, so we speculate the reason may be the Chinese milk vetch and ploughing down milk vetch could promote the degradation of paclobutrazol residue by soil microorganisms, the specific reasons need to be further explored.

Comment 3: Please mention one format of unit i.e. /L or l-1 for all the unit types.

Response: We thank for your careful check and have accepted this suggestion. The format of unit "/L " has been replaced by " l-1" for all the unit types in the revised manuscript.

Comment 4: Sub-heading 3.3 should be corrected as other sub-headings.

Response: We thank for your careful check. The sub-heading 3.3 has been corrected as other sub-headings in the revised manuscript.

Comment 5: Line 129: The authors indicate that "….which away from…." the phrase should be: "….which was away from….".

Response: We thank for your careful check and have accepted this suggestion. This phrase has been changed to “….which was away from….” in line 131 page 6 in the revised manuscript.

Comment 6: Line 146: The authors indicate that "….with extracting solution of…." the phrase should be: "….with an extracting solution of….".

Response: We thank for your careful check. This phrase has been changed to “….with an extracting solution of….” in line 148 page 7 in the revised manuscript.

Comment 7: Line 185: The authors indicate that "….significant…." the word should be: "….significance….".

Response: We thank for your careful check. The word has been revised to “…. significance….” in line 187 page 9 in the revised manuscript.

Comment 8: Line 191: The authors indicate that "….plant…." the word should be: "….plants….".

Response: We thank for your careful check. The word has been revised to “…. plants….” in line 193 page 9 in the revised manuscript.

Comment 9: Line 193: The authors indicate that "….plant…." the word should be: "….plants….".

Response: We thank for your careful check. The word has been revised to “…. plants….” in line 195 page 9 in the revised manuscript.

Comment 10: Line 194: The authors indicate that "….effect…." the word should be: "….effects….".

Response: We thank for your careful check. The word has been revised to “…. effects….” in line 196 page 9 in the revised manuscript.

Comment 11: Line 327: The authors indicate that "….flower…." the word should be: "….flowers….".

Response: We thank for your careful check. The word has been changed to “…. flowers….” in line 330 page 16 and modified in all text in the revised manuscript. 

Comment 12: Line 337: The authors indicate that "….develop…." the word should be: "….developing….".

Response: We thank for your careful check. The word has been changed to “…. developing….” in line 340 page 16 and modified in all text in the revised manuscript.

Other changes: 

1. In the revised manuscript, we added an author “Wei Ren”who was contributed a lot to the analysis of experimental data, and added this author’s list and affiliations and contribution.

Yours sincerely,

Chunzeng Liu

---

## [Decision Letter · Decision Letter 1]

2 Jan 2021

Flower and pod development, grain-setting characteristics and grain yield in Chinese milk vetch (Astragalus sinicus L.) in response to pre-anthesis foliar application of paclobutrazol

PONE-D-20-26606R1

Dear Dr. Liu

We’re pleased to inform you that your manuscript has been judged scientifically suitable for publication and will be formally accepted for publication once it meets all outstanding technical requirements.

Kind regards,

Vijay Gahlaut, Ph.D.

Academic Editor

PLOS ONE

Additional Editor Comments (optional): NIL

Reviewers' comments:

Reviewer's Responses to Questions

**Comments to the Author**

1. If the authors have adequately addressed your comments raised in a previous round of review and you feel that this manuscript is now acceptable for publication, you may indicate that here to bypass the “Comments to the Author” section, enter your conflict of interest statement in the “Confidential to Editor” section, and submit your "Accept" recommendation.

Reviewer #1: All comments have been addressed

Reviewer #2: All comments have been addressed

2. Is the manuscript technically sound, and do the data support the conclusions?

Reviewer #1: Yes

Reviewer #2: Yes

3. Has the statistical analysis been performed appropriately and rigorously? 

Reviewer #1: Yes

Reviewer #2: Yes

4. Have the authors made all data underlying the findings in their manuscript fully available?

Reviewer #1: Yes

Reviewer #2: Yes

5. Is the manuscript presented in an intelligible fashion and written in standard English?

Reviewer #1: Yes

Reviewer #2: Yes

6. Review Comments to the Author

Reviewer #1: No further modifications are required. Manuscript is well-written and the research has wider scopes in agriculture industry.

Reviewer #2: (No Response)

7. PLOS authors have the option to publish the peer review history of their article (what does this mean?). If published, this will include your full peer review and any attached files.

Reviewer #1: **Yes: **Arpan Modi

Reviewer #2: No

---

## [Editor Report · Acceptance letter]

28 Jan 2021

PONE-D-20-26606R1 

Flower and pod development, grain-setting characteristics and grain yield in Chinese milk vetch (*Astragalus sinicus* L.) in response to pre-anthesis foliar application of paclobutrazol 

Dear Dr. Liu:

I'm pleased to inform you that your manuscript has been deemed suitable for publication in PLOS ONE. Congratulations! Your manuscript is now with our production department. 

Kind regards, 

on behalf of

Dr. Vijay Gahlaut 

Academic Editor

PLOS ONE